# Cross-Cultural Adaptation and Patient Evaluations of a Question Prompt List for Danish-Speaking Patients at the Orthopaedic Surgery Department, Odense University Hospital

**DOI:** 10.3390/ijerph22030399

**Published:** 2025-03-10

**Authors:** Nina Høy Chodkiewicz, Emma Weiss Christensen, Meg Chiswell, Peter Martin, Christina Louise Lindhardt

**Affiliations:** 1Clinical Institute, Faculty of Health Science, University of Southern Denmark, 5230 Odense, Denmark; nina.hoy@outlook.dk (N.H.C.); emld@rsyd.dk (E.W.C.); 2Centre for Research in Patient Communication (CFPK), Clinical Development, Odense University Hospital, 5000 Odense, Denmark; 3School of Medicine, Deakin University, Geelong, VIC 3217, Australia; m.chiswell@deakin.edu.au (M.C.); peter.martin@deakin.edu.au (P.M.)

**Keywords:** Question Prompt List, QPL, cross-cultural adaptation, patient-centred communication, health literacy, 5-point Likert scale, semi-structured interviews

## Abstract

Objectives: The Question Prompt List (QPL) enhances patients’ consultation preparation by improving health literacy and communication with healthcare professionals. A validated tool for this context is not yet available. We adapted and translated an Australian QPL to ensure its validity. This study seeks to improve patient-centred communication and health literacy through the QPL. This study aims to conduct a translation and cross-cultural adaptation of the QPL. Study design: Cross-cultural adaptation. Methods: Beaton’s five-step guide inspired the cross-cultural adaptation, which involved forward and back translations from the Australian to Danish versions. Patients at the orthopaedic department tested the final version, evaluating the QPL for clarity, usefulness, and context. Four semi-structured interviews supplemented this process. Feedback from patients led to the refinement and finalisation of the QPL. Results: The translations revealed some contradictions, indicating a need for QPL adjustments to fit the Danish context. Most patients were satisfied with the QPL, finding it beneficial and comprehensive. They noted that the QPL would have been helpful in previous consultations. Minor criticisms of specific questions were also raised, leading to further discussion and refinement. Conclusions: This cross-cultural adaptation has ensured the validity and quality of a Danish QPL, and implementation strategies are now ready for investigation.

## 1. Introduction

Person-centred care (PCC) has gained increasing attention in the literature since the late 1960s, and today, it is still highly relevant to obtain a focus on the patient [1]. Shared decision-making (SDM) is a fundamental aspect of person-centred care (PCC), involving collaboration between patients and healthcare professionals in the decision-making process. SDM facilitates the exchange of information, ensuring that patients are equipped to make informed choices while establishing a mutual understanding [2]. Engaging patients in healthcare decisions has multiple benefits, including improved knowledge, enhanced perception of risk, and better decision-making outcomes. For healthcare professionals, SDM provides access to evidence-based resources, while for healthcare systems, it contributes to improved safety, quality of care, and reduced variability in treatment [3]. The recent literature emphasises the importance of addressing the technical aspects of patient-centred communication, including using a Question Prompt List (QPL), to enhance healthcare delivery and improve patient outcomes [4,5,6,7].

Practical communication skills are essential for providing quality care and treatment. Prior research has demonstrated that patient-centred communication positively impacts essential health outcomes such as pain and stress reduction, less discomfort, and higher quality of life [8,9,10]. Patient-centred communication implies that healthcare professionals should not limit the patient to being just a patient during individual visits but treat the patient as a whole [8,9,10]. Studies have shown that patient education helps patients manage their treatment and care [11,12]. Further, improving patients’ health literacy may enhance communication between healthcare professionals and patients [13,14]. A QPL is crucial in promoting health literacy among patients [12]. Health literacy refers to obtaining, processing, and understanding basic health information and services necessary to make appropriate health decisions. A QPL provides patients with a structured framework to engage with healthcare information, empowering them to ask informed questions and seek clarification on complex medical concepts. Using qualified tools like a QPL is a way to ensure patient-centred communication and improve health literacy. A QPL has positively impacted patients’ communication skills and health literacy [14,15].

This QPL is a set of questions designed to help patients ask relevant questions during their interactions with healthcare professionals. Patients receive the list before their consultations, allowing them to review and highlight any questions they may have before the consultation. According to research in other European countries, using QPL positively impacts patient-centred communication and health literacy [3,15,16,17,18,19]. Therefore, this article has an evidence-based foundation for why it would benefit patient-centred communication and health literacy. This study was conducted in collaboration with the Centre for Research in Patient Communication and the Centre for Organisational Change in Person-centred Healthcare. It was conducted in an orthopaedic surgery department at OUH.

The orthopaedics field is complex and covers many areas. Despite the treatment and care provided by orthopaedic surgery, patient complaints are multiple [20]. Further, orthopaedic surgery is skilled craftsmanship; historically, communication between patients and doctors has not been high on the agenda [3]. Therefore, a QPL is much in favour of the patients and the doctors treating the patients. In an initial contact with the orthopaedic ward, the researchers asked five independent doctors how a QPL could be helpful in their contact with the patients. All the doctors replied that the QPL tool would support communication with the patients during the first encounter in the ward. Increasing the patient’s health literacy in an orthopaedic Surgery Department is highly relevant because it is a typical department to overlook the patient [20].

The healthcare systems in Denmark and Australia are influenced by their unique cultural contexts, which affect the organisation, accessibility, and patient experience. Denmark’s system is publicly funded and emphasises equality and patient-centred care, while Australia’s system combines public and private financing, providing flexibility but potentially varying care quality [21,22]. Cultural differences impact patient access and experiences, highlighting the importance of considering these factors when transferring tools like the QPL between countries [23]. Given the relative similarity of the two healthcare systems, it may be necessary to adapt the Australian QPL to the Danish context [21]. This study has been conducted as part of a master’s thesis focusing on ensuring validation through a cross-cultural adaptation and translation of this QPL. This study differs from others by including a topic with limited current data. The QPL aims to increase the patient’s health literacy and improve patient-centred communication between the patient and health professionals.

## 2. Methods

This study used qualitative interviews to respond to the study aim, and COREQ guidelines were used to plan, structure, and report this study. The cross-cultural adaptation was guided by Beaton’s five-step approach, involving a structured forward and backward translation process from the original Australian version to Danish.

### 2.1. The Question Prompt List

The QPL, initially developed by the Centre for Organisational Change in Person-Centred Healthcare (OCPH) at Deakin University, Australia, consists of 36 questions divided into subtopics: “My Illness”, “My Treatment”, “My Tests”, and “My Care” on one side, and “Ideas”, “Concerns”, and “Feelings” on the other. It was designed for a department treating cancer patients, providing a structured approach to patient-centred communication. Permission for translation and adaptation was obtained from its creators. The QPL was administered by two authors of this article at the Orthopaedic Surgery Department at Odense University Hospital in 2024, ensuring its application aligns with its intended purpose.

### 2.2. Cross-Cultural Adaptation

Ensuring validity requires more than translation; specific guidelines are essential. However, there is a lack of clear guidelines for adapting QPL. We followed Beaton et al. ‘s five recommended steps to address this, providing a systematic approach for adapting self-reported measures [23]. This method was chosen because it is widely recognised in cross-cultural adaptation research and ensures linguistic and conceptual equivalence. Beaton’s framework is particularly suited for QPL, as it systematically accounts for semantic, idiomatic, experiential, and conceptual equivalence—key factors ensuring the tool remains effective in a new cultural context. The Danish translation of the QPL is based on the Australian version from OPCH Deakin University.

Stage 1—Initial translation: This translation began with a machine-based translation (AI) (Translator 1), followed by corrections from a translator with a health science background (Translator 2). Two co-authors conducted a detailed review to ensure understanding of the Question Prompt List (Translator 3). As recommended by Beaton et al. [23], the authors had varied backgrounds: Translator 1 was AI for a quick starting point, Translator 2 had a health science background, and Translator 3, a co-author, had a comprehensive understanding of the study’s objectives.

Stage 2—Discussion of the Danish version: Translators 2 and 3 agreed on the final translation, ensuring a common consensus between the two versions.

Stage 3—Back translation: The back translation involved AI (B1) for the initial English version. Three individuals then assessed the B1 rendition: one native English speaker (B2) and two native Danish speakers (B3 and B4) who regularly communicate in English daily.

Stage 4—Expert Committee: A committee reviewed all translated versions against the original QPL. After extensive deliberation, the pre-final iteration was developed. The panel included a back-translator three and Translators 2 and 3. The pre-final QPL adhered to Beaton et al. ‘s guidelines [23,24]. Each query was thoroughly examined from various perspectives, emphasising identifying semantic discrepancies, linguistic nuances, experiential variations and conceptual disparities.

Stage 5—The final version of the QPL was developed as a comprehensive tool and pilot-tested among patients awaiting consultations at the Orthopaedic Surgery Department. Fourteen patients evaluated the QPL questions using a 5-point Likert scale and provided feedback. Four semi-structured telephone interviews followed.

## 3. Results

In the process of a cross-cultural adaptation, 36 questions were reviewed, and modifications were made due to the cultural differences in Denmark. The five stages follow the guidelines from the literature with a few modifications [23]. A few changes were made to the original version in stages two and four. In stage two, a grammatical adjustment was made, and a question underwent revision to suit the context better—specifically, the modified question pertained to the payment associated with each treatment. In Denmark, the healthcare system operates differently, with patients not being required to pay for hospital treatments. Instead, healthcare expenses are covered through taxation [21]. Therefore, this question needs to be more relevant in a Danish context. However, there are specific services that require a user’s fee. Hence, the question was changed to “What kind of services do I need to pay for?”.

In stage four, changes were made in the text around the 36 main questions. The purpose of this change was to make the text more descriptive and accessible for the patient to understand. Also, these changes were made to comply with the limited time for each consultation. The expert committee agreed upon these changes. Overall, the expert committee aimed to preserve the essence of the original QPL while incorporating only those modifications necessary to address cultural variations.

### 3.1. Semi-Structured Interviews

A thematic analysis of the four semi-structured interviews revealed four main themes: the QPL’s applicability (1), the significance of included questions (2), modifications to QPL queries (3), and patient experiences (4).

(1)Patients prepare before their consultation, during which the Question Prompt List is found to be pertinent and suitable for addressing their requirements.(2)Stage five occurred within an orthopaedic department, to patients primarily with injuries rather than illnesses. Specific questions were deemed irrelevant to the patient’s circumstances because they mainly contended with injuries rather than diseases.(3)Each patient has different approaches and abilities to handle each consultation. Therefore, their needs for each question may differ. The questions that are pointed out concern what services and supports are available (question 18), the payment of unique treatments (question 12), and the side effects. One of the informants also suggested a change in the title.(4)The four patients’ approaches to using the Question Prompt List varied based on their previous experiences. They recognised its significance, especially for patients with limited experience in the Danish healthcare system. They agreed on the importance of such tools in guiding patients through consultations to address their healthcare needs.

### 3.2. Five-Point Likert Scale

Fifteen patients participated in the assessment using a five-point Likert scale, with 53% identifying as male. The median age of participants was 44 years, ranging from 19 to 75. Each question was evaluated based on clarity, usefulness, and relevance to the clinical context.

Histograms were created for each category, grouping responses into three levels: low (Options 1 and 2, represented in dark and light purple), medium (Option 3, pink), and high (Options 4 and 5, dark and light green) (Figure 1, Figure 2 and Figure 3). The *X*-axis represents the 36 QPL questions, while the *Y*-axis indicates the percentage of responses for each category. Questions with less than 50% high ratings were considered low-scoring and subjected to further review. Due to the limited number of responses, a statistical analysis could not be performed.

#### 3.2.1. Clarity of Words

Most of the questions received a score of >80% in the categories “very understandable” and “partially understandable” in most questions regarding word choice. Therefore, there is a generally good understanding of all questions in the QPL (Figure 1).

#### 3.2.2. Usefulness

Figure 2 shows the variation in the usefulness of the 36 QPL questions. Questions 12, 23, 31, and 32 scored below 50% for very or partially usable: “Do I have to pay for any of this myself?” “What do I think caused my problem?” “How will others react or be affected?” and “How do I tell my family about my condition?” Question 4, “Can I pass it on to somebody else in my family?” received mixed feedback, with 40% finding it unhelpful. In contrast, questions 10 and 13, addressing “How helpful can I expect the treatment to be?” and “What are the results of the test I’ve had?” received high scores (100%), indicating they are very or partially usable.

#### 3.2.3. Context

There is variation in the context of the 36 QPL questions (Figure 3). Questions 3, 4, 12, 21, 22, 23, 31, and 32 scored 50% or less for being partially or not usable. These include: “How common is it?”, “Can I pass it to somebody else in my family?” “Do I have to pay for any of this myself?” “Who can I talk to for support?” “Who can my family or friends talk to?” “What do I think caused my problem?” “How will others react or be affected?” and “How do I tell my family about my condition?”. In contrast, question 10, “How helpful can I expect the treatment to be?” received a score of 100% in strongly agree or agree, indicating its relevance (Figure 3).

Although some questions in the category’s usefulness and context scored below 50%, we have kept them from the QPL. Because this QPL was developed to be a general tool for all departments at the hospital, we have decided to keep the questions as they may be relevant for other departments and patients.

## 4. Discussion

Using a QPL in patient communication has been identified as a promising strategy to improve patient engagement, satisfaction, and understanding in healthcare settings. Our study examined the cultural differences and explored the relevance of the QPL as a patient-centred tool.

Our research undermines the positive impact that a QPL can have on facilitating more comprehensive and patient-centred dialogues between healthcare providers and patients [9,10,25]. QPL provide structured prompts that enable patients to express their concerns, preferences, and information needs more effectively during medical consultations. This enhances collaboration between healthcare providers and patients, leading to a more empowered and engaged patient population in their care decision-making processes. One of the notable advantages highlighted by our research is the role of a QPL in addressing communication barriers commonly encountered in healthcare encounters [2]. A QPL serves as a valuable tool to mitigate these challenges by offering a structured framework for patients to organise their thoughts and prioritise their questions, thereby enhancing the clarity and efficiency of communication. A limitation of utilising this Quality of Patient Life (QPL) tool is the uncertainty regarding its impact on the time healthcare staff requires to conduct consultations. Consequently, exploring this factor within an implementation plan is essential to assess the potential time used by the healthcare staff.

Implementing the QPL requires several factors to be upheld. First, the management team on the orthopaedic ward must support the implementation, and second, it must be introduced thoroughly to the staff on the ward who will use it in the daily clinic. The QPL could, for example, be tested and introduced on a smaller number of patients and fully implemented on all the patients in the ward after 12 weeks. This may ease the implementation as nurses and doctors would be trained in small groups, simultaneously avoiding all the staff being away from the treatment and care [26].

In recent studies, the adaptation and implementation of Question Prompt Lists (QPLs) across different cultural and healthcare settings have significantly impacted patient engagement and information-seeking behaviour. One study focused on adapting an Australian QPL for oncology to a Norwegian setting, applying a combined method approach to ensure cultural and contextual relevance [3]. The findings emphasised the importance of tailoring QPLs to local nuances while maintaining the core principles of enhancing patient communication. Similarly, research on non-specialist palliative care revealed that preparing QPLs with input from patients, families, and clinicians could provide valuable insights into their unique needs, helping to bridge gaps in understanding [26]. Furthermore, a longitudinal qualitative study examining the role of QPLs in supporting patients highlighted how endorsing question-asking practices enables patients to acquire the information they seek, fostering a more informed and empowered approach to their care [17]. These findings emphasise the necessity of tailoring QPLs to align with the specific needs of patients and healthcare professionals while considering cultural nuances that influence communication and decision-making. In a Danish context, adapting and implementing a QPL could enhance patient engagement, improve shared decision-making, and support structured communication in clinical consultations, particularly in specialised care settings such as Orthopaedic surgery.

Based on the guidelines by Beaton et al. [23], we could adapt an Australian Question Prompt List (QPL) to a Danish context. The adaptation was feasible due to the cultural similarities between Denmark and Australia. Following Beaton et al. guidelines helped us ensure that the validity of the Danish version of QPL is equivalent to the original Australian version. Our main goal was to make this QPL understandable and beneficial for patients at the Orthopaedic Department and potentially for other departments. The diversity of the expert panel has been crucial to achieving different points of view and making the best possible adaptation. The changes made in each step have been essential to ensure the contextual and cultural differences that may occur in different countries.

The results from the patients in stage five also contributed to gaining knowledge about the patients’ views and perspectives regarding the QPL. Their perspectives and insight have taught us about the cultural changes that can occur in a new county and a different hospital department. Therefore, we have decided to keep some of their suggested changes since this Question Prompt List was developed to be a general tool for all departments at the hospital. Thus, some questions may be more relevant to other patient groups or departments. Also, the patients’ need for each question can differ regarding the type of consultation. This decision also signifies the need for this QPL to be tested in other departments. Tracy et al. argue that a well-designed and easily useable QPL can help patients as healthcare professionals to ask the right questions and thus make the transition in treatment and care more manageable for patients, e.g., in orthopaedic surgery [26]. The results from stage five align with those of another study, in which similar results were found regarding the usefulness and clarity of a QPL [27]. It is a strength that this study includes a variation in patients across various age groups from 19 to 75 years. Thereby, the results of this study are based on different aspects of this QPL. Based on the results, it has been possible to adapt this QPL so that the patients can be more prepared for the consultation. It intends to increase patients’ health literacy and the communication between patients and the healthcare staff in the long term [3].

This QPL is intended to be a general tool that patients can use in every department in the hospital. The QPL must be tested in several departments and with different patient groups to ensure it can be generalised. It should be noted that the results cannot be transferred to other departments or patient groups without looking at the patient category for the specific ward. This can cause variation and may also indicate that the patients have different needs for health skills for their consultations. In conclusion, our study shows that this Question Prompt List has excellent potential for facilitating effective communication between patients and healthcare professionals in treatment and care. The QPL provides a better starting point for patients who may become overwhelmed in consultation while also offering the healthcare staff an opportunity to address key areas critical to the patient. This list can help structure conversations, improve health literacy, and empower patients to participate actively in their healthcare journey. By using a QPL to promote collaborative and patient-centred care, healthcare organisations can aim to improve patient experiences and achieve better health outcomes.

## Figures and Tables

**Figure 1 ijerph-22-00399-f001:**
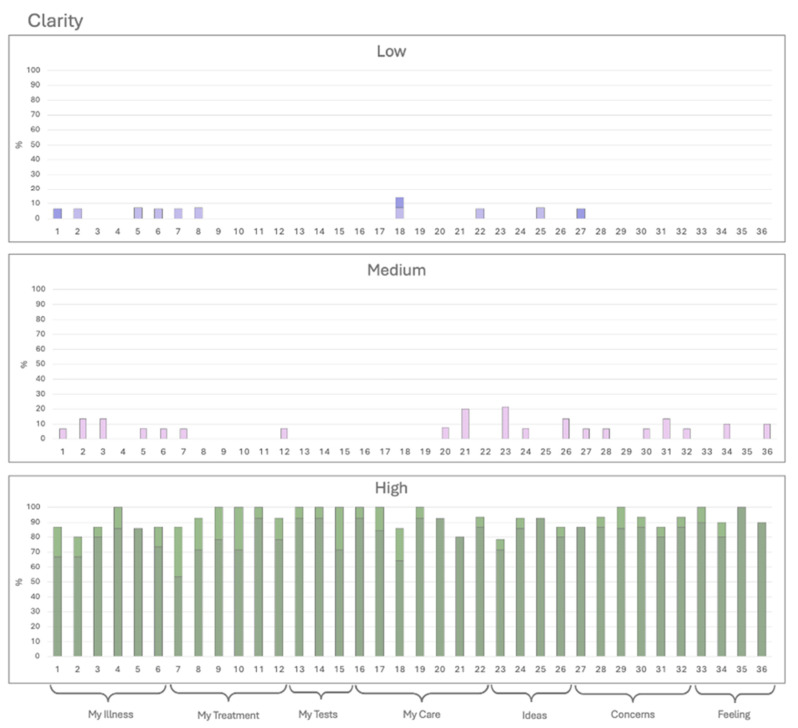
Clarity: The patients rated the clarity of each question on a 5-point Likert scale. Option 1 (dark purple) and 2 (light purple) are expressed on the low end, Option 3 (pink) on medium, and Option 4 (light green) and Option 5 (dark green) on high.

**Figure 2 ijerph-22-00399-f002:**
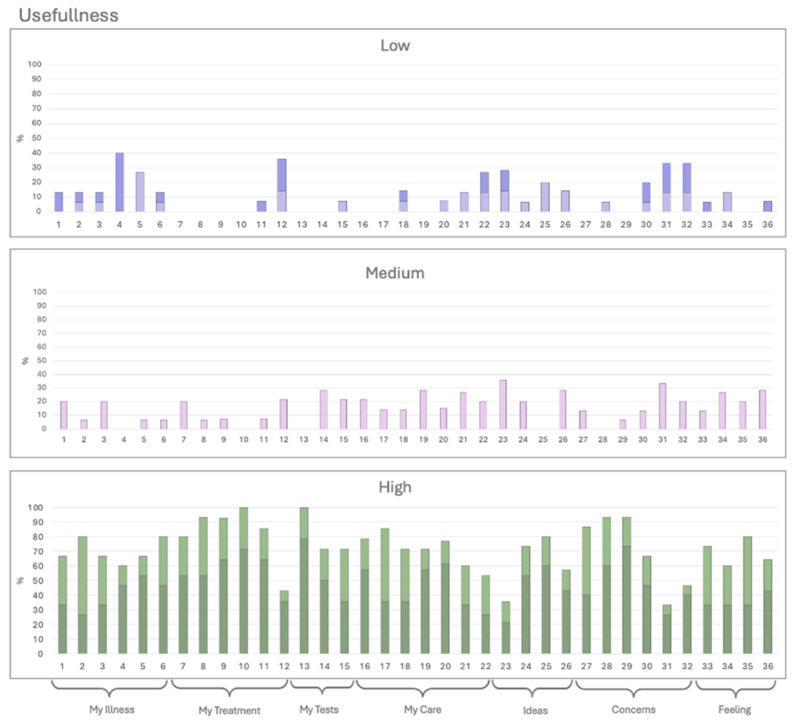
Usefulness. The patients rated the usefulness of each question on a 5-point Likert scale. Option 1 (dark purple) and 2 (light purple) are expressed on the low, option 3 (pink) on medium, and option 4 (light green) and 5 (dark green) on high.

**Figure 3 ijerph-22-00399-f003:**
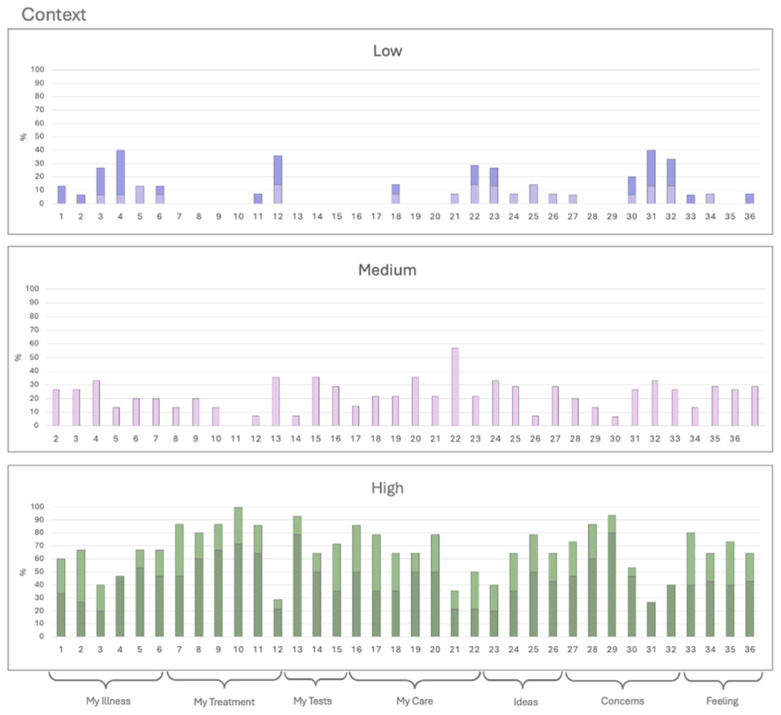
Context. The patients rank the relevance of the context with each question on a 5-point Likert scale. Option 1 (dark purple) and 2 (light purple) are expressed on the low, option 3 (pink) in medium, and option 4 (light green) and 5 (dark green) in high.

## Data Availability

Data can be requested by emailing the first author. N.H.C.

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
