# Peer review of "Cross-Cultural Adaptation and Patient Evaluations of a Question Prompt List for Danish-Speaking Patients at the Orthopaedic Surgery Department, Odense University Hospital"

_ijerph, 2025, doi:10.3390/ijerph22030399_

Round 1

Reviewer 1 Report

Comments and Suggestions for Authors

Communication has always been one of the most critical aspects of quality and safety in healthcare. Today, patients want to be more informed and aware of the type of care they receive. The doctor-patient relationship has changed significantly, as has public opinion toward science. As a result, communication now plays a more crucial role than ever—not only in informing but also in clarifying and persuading.

A recent Lancet editorial reaffirmed the importance of patient involvement in the care process, emphasizing that effective communication is the foundation of everything.
Helen Haskell - Surgical adverse events in the US, BMJ: first published as 10.1136/bmj.q2437 on 13 November 2024.

This premise highlights that the work conducted is valuable and aligned with this perspective.

However, in reviewing the article, I believe several aspects require further clarifications and deeper analysis to better justify the study's purpose.

Examining the various sections, starting with the introduction, the first question that arises is why, in the field of orthopedics—one of the most exposed to compensation claims—the need for tools that help patients better understand the complexities of certain procedures has never been strongly felt. It is essential to understand the cultural barriers that have hindered communication so far, which are crucial for the successful implementation of these tools. For example:

How much time does using QPL take away from surgeons?
How does QPL relate to the informed consent document?
Is the informed consent document truly understandable to patients, or is it primarily a product of defensive medicine?
In the introduction, where numerous articles are cited (line 57), the claim that this study is the only one presenting data should be supported by additional brief comments on the types of research conducted and their findings.

The introduction should be expanded with additional bibliographic references, better justifying the study’s purpose and explaining how QPL integrates with the informed consent document. Additionally, it should clarify how the QPL is administered (by whom, where, and when).

Regarding the methodology, the article refers to Beaton's five-step guide, but the choice of this tool should be better justified. Furthermore, in the discussion, it would be useful to evaluate its advantages and limitations. The collected data is presented as descriptive statistics, but a more detailed description of the sample tested with the QPL would be valuable. For example, it would be helpful to know if there are significant differences based on patients’ educational levels.

The discussion should also include:

Advantages and limitations of these tools
Possible implementation methods for the QPL
Contextual factors influencing its use
Training needs for healthcare providers
Time required for implementation

Conclusion
The topic is highly relevant, but the study requires further elaboration in the introduction to better define its objectives. The methodology and data analysis sections are overly concise, and the discussion should be expanded further.

Author Response

Response to reviewer #1 on manuscript: Manuscript ID: ijerph-3486448

Ad 1. Reviewer #1: Communication has always been one of the most critical aspects of quality and safety in healthcare. Today, patients want to be more informed and aware of the type of care they receive. The doctor-patient relationship has changed significantly, as has public opinion toward science. As a result, communication now plays a more crucial role than ever—not only in informing but also in clarifying and persuading.

Response: Thank you for your comment

Ad 2.  A recent Lancet editorial reaffirmed the importance of patient involvement in the care process, emphasizing that effective communication is the foundation of everything.
Helen Haskell - Surgical adverse events in the US, BMJ: first published as 10.1136/bmj.q2437 on 13 November 2024. This premise highlights that the work is valuable and aligned with this perspective.

Response: Thank you for this suggestion; we have included it in the introduction.

Ad 3. However, in reviewing the article, I believe several aspects require further clarification and deeper analysis to justify the study's purpose better.

Examining the various sections, starting with the introduction, the first question that arises is why, in orthopedics—one of the most exposed to compensation claims—the need for tools that help patients better understand the complexities of specific procedures has never been strongly felt. It is essential to understand the cultural barriers that have hindered communication so far, which are crucial for the successful implementation of these tools. For example:

How much time does using QPL take away from surgeons?
How does QPL relate to the informed consent document?

Response: We acknowledge your comments and will address the suggestions to improve the article and make it ready for publication.

Ad 3. Is the informed consent document truly understandable to patients, or is it primarily a product of defensive medicine?

Response: We acknowledge the comment and we can inform you that the consent form has been reviewed and approved following the Danish guidelines. The informed consent document is specific to this research and does not extend beyond it. Patients will not receive an informed consent document regarding the QPL in the future.

Ad 4. In the introduction, where numerous articles are cited (line 57), the claim that this study is the only one presenting data should be supported by additional brief comments on the types of research conducted and their findings.

The introduction should be expanded with additional bibliographic references, better justifying the study’s purpose and explaining how QPL integrates with the informed consent document. Additionally, it should clarify how the QPL is administered (by whom, where, and when).

Response: Thank you for your valuable suggestions. We agree that the claim about this study being the only one presenting data should be supported with more context on the types of research and their findings. We will expand the introduction with additional bibliographic references to better justify the study’s purpose and clarify how the QPL integrates with the informed consent document. We will also provide more details on how the QPL is administered, including who administers it, where, and when.

Ad 5. Regarding the methodology, the article refers to Beaton's five-step guide, but the choice of this tool should be better justified. Furthermore, in the discussion, it would be useful to evaluate its advantages and limitations. The collected data is presented as descriptive statistics, but a more detailed description of the sample tested with the QPL would be valuable. For example, it would be helpful to know if there are significant differences based on patients’ educational levels.

The discussion should also include:

Advantages and limitations of these tools
Possible implementation methods for the QPL
Contextual factors influencing its use
Training needs for healthcare providers
Time required for implementation

Response: Thank you for your feedback. In the methodology section, we will address the need for a stronger justification for using Beaton’s five-step guide. Additionally, we will expand the discussion to include an evaluation of its advantages and limitations, explore potential differences in the sample based on educational levels, and incorporate details on implementation methods, contextual factors, healthcare provider training, and time requirements. Your suggestions will help strengthen the article.

Ad 6. Conclusion
The topic is highly relevant, but the study requires further elaboration in the introduction to better define its objectives. The methodology and data analysis sections are overly concise, and the discussion should be expanded further.

Response: Thank you for your feedback. We agree that further elaboration in the introduction would help clarify the study's objectives. We will also work on expanding the methodology, data analysis, and discussion sections to provide more detail and depth.

Reviewer 2 Report

Comments and Suggestions for Authors

This article aims to translate and cross-culturally adapt the Question Prompt List from the original Australian version into the Danish culture. It presents the different phases of this methodological study, contributing to its use in this culture.

The method and results need to be reformulated, some subchapters are not suitable (example: 5-point Likert scale)

Abstract

Review objective. The aim is the translation and cross-cultural adaptation of the QPL.

Study design: is it a methodological study of translation and cross-cultural adaptation?

Results: there are several types of validity, what type of validity do the authors refer to?

1.Introduction

Line 70 and 71: I suggest reviewing the objective, as stated in the abstract.

2.Methods

In this chapter it is important to mention the type of study and identify the author who serves as support for its development.

2.1. The Question Prompt List: refer to the QPL answer type (5-Point Likert Scale)

2.2. Cross-Cultural Adaptation

I suggest presenting the original questions, translated version, back-translated version, translated and culturally adapted version and final version in a table (supplementary document).

3.Results

The results are presented in an unclear way

4.Discussion

Some statements at the beginning of the discussion are not related to the objectives of the study, namely: In our study, we examined the impact of a QPL on patient-provider interactions and explored their effectiveness and implications for clinical practice.

Author Response

Response to reviewer #2 on manuscript: Manuscript ID: ijerph-3486448

Ad 1. Reviewer #2: This article aims to translate and cross-culturally adapt the Question Prompt List from the original Australian version into the Danish culture. It presents the different phases of this methodological study, contributing to its use in this culture. The method and results need to be reformulated, some subchapters are not suitable (example: 5-point Likert scale)

Response: Thank you for your feedback. We will revise the method and results sections to improve clarity and ensure alignment with the study’s objectives. Additionally, we will review the subchapters and make necessary adjustments. We appreciate your suggestions and will refine the text accordingly

Ad 2. Abstract: Review objective. The aim is the translation and cross-cultural adaptation of the QPL.

Study design: is it a methodological study of translation and cross-cultural adaptation?

Response: Thank you for your suggestion. We have ensured that the abstract and aim clearly state the objective.

Ad 3. Results: there are several types of validity, what type of validity do the authors refer to?

Response: Thank you for your comment. We have ensured validity throughout the entire process, following the relevant theoretical framework to ensure a consistent and reliable approach.

Ad 4. 1.Introduction

Line 70 and 71: I suggest reviewing the objective, as stated in the abstract.

Response: Thank you for your suggestion. We will review the objective in lines 70 and 71 to ensure alignment with the abstract and make any necessary revisions. We appreciate your feedback.

Ad 5. 2.Methods

In this chapter it is important to mention the type of study and identify the author who serves as support for its development.

2.1. The Question Prompt List: refer to the QPL answer type (5-Point Likert Scale)

2.2. Cross-Cultural Adaptation

I suggest presenting the original questions, translated version, back-translated version, translated and culturally adapted version and final version in a table (supplementary document).

Response: Thank you for your valuable suggestions. We will ensure that the type of study is clearly mentioned, along with the identification of the author supporting its development.

Ad 6. 3.Results

The results are presented in an unclear way

Response: We have changed the results section to present the findings more clearly.

Ad 7. 4.Discussion

Some statements at the beginning of the discussion are not related to the objectives of the study, namely: In our study, we examined the impact of a QPL on patient-provider interactions and explored their effectiveness and implications for clinical practice.

Response: We have been through the text in the discussion section, and we hope You find the section improved and to Your satisfaction.

Round 2

Reviewer 1 Report

Comments and Suggestions for Authors

Thank you for your new and comprehensive revision of the paper. The authors have adequately addressed the considerations and comments. The paper is suitable for publication in its current form.

Reviewer 2 Report

Comments and Suggestions for Authors

Thank you very much for changing the manuscript according to the suggestions.
It became clearer and with greater methodological rigor.